# Partial Substitution of Urea with Biochar Induced Improvements in Soil Enzymes Activity, Ammonia-Nitrite Oxidizers, and Nitrogen Uptake in the Double-Cropping Rice System

**DOI:** 10.3390/microorganisms11020527

**Published:** 2023-02-19

**Authors:** Saif Ullah, Izhar Ali, Mei Yang, Quan Zhao, Anas Iqbal, Xiaoyan Wu, Shakeel Ahmad, Ihsan Muhammad, Abdullah Khan, Muhammad Adnan, Pengli Yuan, Ligeng Jiang

**Affiliations:** 1Key Laboratory of Crop Cultivation and Physiology, Guangxi University, Education Department of Guangxi, Nanning 530004, China; 2Guangxi Key Laboratory of Forest Ecology and Conservation, College of Forestry, Guangxi University, Nanning 530004, China; 3Guangxi Key Laboratory of Agro-Environment and Agro-Products Safety, College of Agriculture, Guangxi University, Nanning 530004, China

**Keywords:** rice, biochar, soil organic carbon, nitrogen, soil enzymes, ammonia-oxidizing bacteria

## Abstract

Biochar is an important soil amendment that can enhance the biological properties of soil, as well as nitrogen (N) uptake and utilization in N-fertilized crops. However, few studies have characterized the effects of urea and biochar application on soil biochemical traits and its effect on paddy rice. Therefore, a field trial was conducted in the early and late seasons of 2020 in a randomized complete block design with two N levels (135 and 180 kg ha^−1^) and four levels of biochar (0, 10, 20, and 30 t ha^−1^). The treatment combinations were as follows: 135 kg N ha^−1^ + 0 t B ha^−1^ (T1), 135 kg N ha^−1^ + 10 t B ha^−1^ (T2), 135 kg N ha^−1^ + 20 t B ha^−1^ (T3), 135 kg N ha^−1^ + 30 t B ha^−1^ (T4), 180 kg N ha^−1^ + 0 t B ha^−1^ (T5), 180 kg N ha^−1^ + 10 t B ha^−1^ (T6), 180 kg N ha^−1^ + 20 t B ha^−1^ (T7) and 180 kg N ha^−1^ + 30 t B ha^−1^ (T8). The results showed that soil amended with biochar had higher soil pH, soil organic carbon content, total nitrogen content, and mineral nitrogen (NH_4_^+^-N and NO_3_^−^-N) than soil that had not been amended with biochar. In both seasons, the 20 t ha^−1^ and 30 t ha^−1^ biochar treatments had the highest an average concentrations of NO_3_^–^-N (10.54 mg kg^−1^ and 10.25 mg kg^−1^, respectively). In comparison to soil that had not been treated with biochar, the average activity of the enzymes urease, polyphenol oxidase, dehydrogenase, and chitinase was, respectively, 25.28%, 14.13%, 67.76%, and 22.26% greater; however, the activity of the enzyme catalase was 15.06% lower in both seasons. Application of biochar considerably increased the abundance of ammonia-oxidizing bacteria (AOB), which was 48% greater on average in biochar-amended soil than in unamended soil. However, there were no significant variations in the abundances of ammonia-oxidizing archaea (AOA) or nitrite-oxidizing bacteria (NOB) across treatments. In comparison to soil that had not been treated with biochar, the average N content was 24.46%, 20.47%, and 19.08% higher in the stem, leaves, and panicles, respectively. In general, adding biochar at a rate of 20 to 30 t ha^−1^ with low-dose urea (135 kg N ha^−1^) is a beneficial technique for improving the nutrient balance and biological processes of soil, as well as the N uptake and grain yield of rice plants.

## 1. Introduction

Rice serves as the main staple diet for more than half of the world’s population and was cultivated on more than 165.25 Mha of land worldwide in 2019–20, and a total of 466.31 Mt of grain was produced annually during this period; losses of nitrogen (N) thus pose a serious challenge to paddy rice cultivation [1,2]. In China, 23% of the country’s croplands are utilized for rice production, representing roughly 20% of the world’s total [3,4]. A meta-analysis revealed that the N use efficiency of rice was 28.1% in China from 2000 to 2005 [5], which is lower compared with estimates of N use efficiency in the United States (52%) and Europe (68%) over the same period [6]. Future generations will be under more strain due to the modern era’s ever-growing demand for food and energy, as fossil fuel use rises and the energy crisis intensifies. To meet these problems and reduce environmental risks, new ecofriendly measures are necessary. The use of environmentally friendly techniques that emit fewer greenhouse gases and mitigate N leaching losses can reduce the ecological costs of farming [7] and maintain the capacity of ecosystems to supply ecosystem services without compromising food security.

The addition of biochar to soil as an organic amendment to increase soil carbon (C) stocks, enhance soil physiochemical and biological properties, accelerate soil nutrient cycling and enhance crop yields has attracted much research attention [8,9,10]. Prior studies have revealed that the biochar in soil can enhance the availability of nutrients and C sequestration, increase soil pH and the cation-exchange capacity, and alter nutrient cycling by regulating soil microbial communities [9,11]. In addition, biochar plays a crucial role in the soil N cycle by decreasing the leaching of inorganic N and nitrous oxide (N_2_O) emissions [12], which promotes organic nitrogen fixation [13]. Furthermore, due to its high surface area and ion exchange capacity, biochar could successfully adsorb NH_4_^+^-N and NO_3_^−^-N ions [14]. Thus, biochar has become increasingly used as a soil conditioner to improve soil quality and enhance productivity.

Soil biological properties have been proposed to be key factors in soil biochemical properties, and the microbiological indicators of soil provide a sensitive indicator of soil health in both aquatic and agro-ecosystems [15]. In general, microbial functions and soil fertility are closely linked, because microorganisms play crucial roles in the mineralization of various natural elements, including carbon, nitrogen and phosphorus (P) [16], and soil enzymes facilitate biochemical processes in the soil [17]. Enzymatic activity in soil is considered to be one of the more sensitive markers of anthropogenic disturbance [18]. Soil enzymatic activity also provides a reflection of the biological properties of soil [2]. The types of enzymes, the amount of biochar applied, and the characteristics of the soil all have a major role in how biochar affects soil enzyme activities. The soil enzymes’ activity rate due to biochar application in the soil largely depends on the enzyme types, the amount of biochar applied and the soil properties [9,19].

Nitrification is one of the crucial process in the N cycle; it involves microorganisms oxidizing ammonium (NH_4_^+^) to produce nitrate (NO_3_^−^), which then releases soil N for plant growth and development [20] and production of a denitrification substrate [21]. According to Li et al. [22], nitrification involves two stages: the oxidation of ammonia to nitrite, which is catalyzed by ammonia-oxidizing bacteria (AOB) and archaea (AOA), and the oxidation of nitrite to nitrate, which is catalyzed by nitrite-oxidizing bacteria (NOB). The majority of these microbes are autotrophic aerobes. Although their particular metabolism gives them a special niche, this in turn causes delayed and ineffective growth. The rate of application of biochar can change soil N_2_O emissions, which in turn can affect the abundances of AOB, AOA, and NOB [23]. Few pieces of research, meanwhile, have examined how biochar rates with high- and low-dose urea application can affect the enzymes activity, AOB, AOA, and NOB abundances in paddy soil [24].

In recent years, there has been a lot of focus on the advantages that biochar has for soil qualities. A few recent studies have shown that biochar has the ability to alter N cycling in soil by affecting nitrification and denitrification, which are a major sources of N_2_O [13,23]. Hardly any investigations have studied the effects of different levels of biochar under high- and low-dose N application on soil chemical properties, soil enzyme activities, and the abundances of AOB, AOA, and NOB in double-cropping rice systems. Due to its unique physical and chemical characteristics, we thus hypothesized that biochar can be utilized as a soil supplement to increase soil nutrients, augment the availability of microorganisms, and increase rice yield. The specific objectives of this study were as follows: (1) to assess whether biochar amendment has a significant effect on soil nutrients or changes the activities of soil enzymes and nitrogen cycling microbes; and (2) to compare whether there is a significant difference in plant nitrogen uptake and the grain yield of rice under low- and high-dose urea application with various biochar rates.

## 2. Materials and Methods

### 2.1. Experimental Site and Crop Management

A field experiment in the early (March–July) and late (August–November) season in 2020 was carried out at the experimental station (22°49′0.01″ N, 108°19′0.01″ E) of Guangxi University. This site experiences a warm, humid subtropical climate with a seasonal monsoon and an annual average temperature and precipitation of 23.29 °C and 1491 mm, respectively (Figure 1). The soil of the experimental field is classified as a Ultisol, and the basic physio-chemical properties are as follows: pH (5.91), bulk density (1.37 g cm^−3^), soil organic carbon (SOC) (9.85 g kg^−1^), total N (1.23 g kg^−1^), potassium (K) (12.13 g kg^−1^), and P (0.62 g kg^−1^). The experiments were carried out in a randomized complete block design with two N levels (135 and 180 kg ha^−1^) and four levels of biochar (0, 10, 20, and 30 t ha^−1^). The treatment combinations were as follows (T: treatment; N: nitrogen; B: biochar): T1: = 135 kg N ha^−1^ + 0 t B ha^−1^; T2: = 135 kg N ha^−1^ + 10 t B ha^−1^; T3: = 135 kg N ha^−1^ + 20 t B ha^−1^; T4: = 135 kg N ha^−1^ + 30 t B ha^−1^; T5: = 180 kg N ha^−1^ + 0 t B ha^−1^; T6: = 180 kg N ha^−1^ + 10 tB ha^−1^; T7: = 180 kg N ha^−1^ + 20 t B ha^−1^; and T8: = 180 kg N ha^−1^ + 30 t B ha^−1^. The experiment’s test crop was the indica inbreed cultivar Zhenguiai, whose grains are commonly used in China to make rice noodles. The size of the plot was 3.9 m × 6 m (23.4 m^2^) and each treatment was repeated three times.

Seeds grown in plastic trays and two seedlings of comparable size were transplanted at 25 days old. The method outlined by Ullah et al. [2] was utilized to prepare the biochar made from cassava straw for the experiment. The used biochar contained 674.00 g kg^−1^ of carbon, 5.43 g kg^−1^ of nitrogen, 48.33 g kg^−1^ of potassium, 46.33 g kg^−1^ of sulfur, and 3.81 g kg^−1^ of hydrogen. The cassava straw had a specific surface area of 2.46 m^2^ g^−1^, an average pore size of C was 3.37 nm, and a C:N ratio of 124.12. Three split applications of urea were made: a basal dose of 50%, a tillering dose of 30%, and a panicle initiation dose of 20%. Table 1 provides thorough information on the timing and rate of application of urea and biochar. Superphosphate was applied at a rate of 75 kg ha^-1^ for phosphorus as a basal dose and potassium chloride for K at a rate of 150 kg ha^−1^, applied in two splits: 50% as a basal dosage and 50% at the tillering stage. From transplantation through physiological maturity, standard flood water was applied to a depth of about 5 cm. Insecticides (chlorantraniliprole formulations sprayed at the approved rate of 150 mL a.i. per ha) and herbicides (paraquat at 24.7 gallons per ha) were also used.

### 2.2. Soil and Plant Sampling

The soil samples from the upper horizon (0–20 cm) in each plot were collected at five randomly selected points from all the replications using a soil auger (3.8 cm in diameter) in both the early and late seasons, soon after harvesting, for measurements of biological and chemical traits. Samples were homogenized in plastic bags, packed in polystyrene boxes with ice inside the boxes, and taken to the laboratory. In the laboratory, a portion of the soil samples was air-dried, ground, and then passed through a 2 mm sieve for analysis of chemical properties; the remaining soil was stored at 4 °C and –80 °C for subsequent determination of the soil enzyme activities and concentrations of AOB, AOA, and NOB, respectively. For measurement of nitrogen accumulation in each part, an equal number of seedlings from 15 hills in each treatment were randomly collected by destructive methods from four border rows on both sides. The samples were divided into stems, leaves, and panicles, kept in an oven at 75 °C for 48 h, and dried until a constant weight was achieved. The separated dried stems, leaves, and spikes were taken to measure nitrogen accumulation in both seasons.

### 2.3. Determination of Soil Chemical Properties and Plant N Concentrations

Digital pH meter (Starter 2100 pH Bench, OHAUS, Parsippany, NJ, USA) was used to record the pH of soil samples after shaking the soil with distilled water at a solid-to-water ratio of 1:2.5 (*w*/*v*) for 1 h. Bao [25] guidelines for measuring soil moisture content were followed. A 0.5 mm sieve was used to filter air-dried soil, and the weight of the tin (g) was recorded as W1. Along with the tin, a 1 g soil sample was collected, and it was weighed as W2. To obtain a constant weight of W3, the soil samples were dried in an oven for two hours at 105 °C. The percentage soil moisture content was measured as
$$MC\;\%=\frac{W2-W3}{W3-W1}$$

Soil organic carbon was measured using the oxidation method. After boiling at 175 °C for 5 min, soil samples (0.5 g) were digested with 5 mL of 1 mol K_2_Cr_2_O_7_ and 5 mL of concentrated H_2_SO_4_. The excess chromic acid is determined by titration with FeSO_4_ and the quantity of the substance oxidized is calculated from the amount of dichromate reduced [25]. Weighing either 1 g of plant tissue or 2 g of soil allowed us to calculate the total nitrogen. 1 g of potassium sulfate, copper sulfate, and selenium powder in a ratio of 100:10:1 were added as a catalyst in a digestion tube along with the sample. Concentrated H_2_SO_4_ was then added (5 mL for the plant samples and 10 mL for the soil). A digester (X20A aluminum module automatic digester, Shanghai Shengsheng Automatic Analytical Instrument Co., Shanghai, China) was used to place the digestion tube, and it was heated to 370 °C before being digested until clear (2 h for plant samples, 4 h for soil samples). The distillate was absorbed with a 2% boric acid solution in the digestion tube after sodium hydroxide (20–30 mL) was added, and the indicator was methyl red bromocresol green. As per Bao. [25], samples were titrated with sulfuric acid, and the volume was recorded to determine the N concentration. Inorganic N (NH_4_^+^-N and NO_3_^−^-N) concentration was assessed using the method described by Maynard et al. [26]. After being shaken for an hour at 200 rpm and filtered through a 0.45-um polysulfone membrane, soil samples were extracted with 2 M KCl. The amount of N that the KCl removed was then quantified colorimetrically.

### 2.4. Soil Enzyme Measurements

Soil urease activity was determined following the method described by Li et al. [27]. Initially, 1 g of soil was placed in a 10-mL centrifuge tube, and 0.2 mL of toluene was added. After 15 min, 2 mL of 10% urea solution and 4 mL of pH 6.7 citrate buffer solution were added; the mixture was then shaken well and incubated in a 37 °C incubator for 24 h. The mixture was centrifuged, and 0.375 mL of the supernatant was added to a 10-mL centrifuge tube. Next, 0.5 mL of sodium phenate solution and 0.375 mL of sodium hypochlorite solution were added, and the mixture was shaken. After 20 min, when the color developed, 5 mL of water was added and the mixture was mixed well. Finally, the urease activity was determined using a spectrophotometer at 578 nm within 1 h. Urease activity was expressed as the number of mg of NH_3_^–^ in 5 g of soil after 24 h. Soil catalase activity was determined following the standard procedure of Li et al. [27]. Specifically, 0.5 g soil samples were placed in a 10-mL volumetric flask, and 6.4 mL of distilled water and 0.8 mL of a 0.3% H_2_O_2_ solution were added; the mixture was then covered tightly and shaken for 20 min, and 0.16 mL of saturated aluminum potash was added, followed by 0.8 mL of 1.5 mol sulfuric acid. After centrifugation, the absorbance of the supernatant was measured at 240 nm; a soil-free control was also used. Catalase activity was expressed in mg of hydrogen peroxide per g of soil decomposed in 20 min.

Polyphenol oxidase activity was determined following the method of Guan et al. [28]. Soil samples (2 g) were placed in a 50-mL Erlenmeyer flask, and then 10 mL of 1% pyrogallol solution was added; the flask was then shaken and kept in a thermostat at 30 °C for 2 h. After removing the mixture from the thermostat, 4 mL of citric acid–phosphate-buffered solution (pH 4.5) was added, followed by 35 mL of ether; the mixture was then shaken vigorously several times, and the extraction procedure was conducted for 30 min. The colored ether phase containing the dissolved purple gallate was colorimetrically determined at a wavelength of 430 nm. To prevent errors caused by ether, the colorimetric solution tank was washed with anhydrous ethanol after every color comparison. To account for the error caused by the original ether-soluble organic matter in the soil and the purity of gallic phenol, soil without substrate and substrate without soil were used as controls. In the control soil without substrate, the substrate was cultivated with water instead of substrate. The amount of purple gallate can be determined from a standard curve drawn with potassium dichromate. The activity of polyphenol oxidase is expressed in mg of purple gallate in 1 g of soil after 2 h.

Dehydrogenase activity was measured by taking 5 g of fresh soil samples that have passed through a 1.25 mm sieve and placing them in a cork with a stopper. Next, 2 mL of 1% 2%, 3%, and 5% triphenyltetrazolium chloride solution and 2 mL of distilled water were added to each of the flasks and mixed thoroughly. They were then placed in a 37 °C incubator in the dark for 6 h. After incubation, 5 mL of methanol was added, and the mixture was shaken vigorously for 1 min, 20 s and left to stand for 5 min, respectively. All the substances in the Erlenmeyer flask were then placed into the colorimetric tube and washed with a small amount of methanol two to three times; the liquid was filtered into the colorimetric tube and diluted to 25 mL, The absorbance value was then measured at 485 nm, as described by Guan et al. [28]. Chitinase activity was determined following the procedure of Bao, [25], weighing 10 g of fresh soil and adding 0.75 mL of toluene. After 15 min, 5 mL of 1% chitin colloid and 10 mL of phosphate buffers (pH 6.0) were added. After incubation at 37 °C for 16 h, the supernatant was centrifuged and frozen. Specifically, 3 mL of enzyme solution was added to 4.5 mL of DNS, which was then boiled in a water bath for 10 min. After cooling, the absorbance was measured at 540 nm. Chitinase activity was equal to the amount of enzyme required to hydrolyze 1 µg of N-acetylglucosamine h^−1^.

### 2.5. DNA Extraction and Quantitative PCR Analysis

Following the manufacturer’s instructions, soil DNA was extracted from 0.5 g of soil using a Fast DNA^®^SPIN Kit (Qbiogen Inc., Carlsbad, CA, USA). On a 1% agarose gel, the extracted DNA was evaluated, and a Nanodrop^®^D-1000 UV-Vis spectrophotometer was used to determine the DNA concentration (NanoDrop Technologies, Wilmington, DE, USA). The PCR analysis employed DNA that had been diluted tenfold. According to Wang et al. [29] the primer pairs utilized are provided in the Appendix A. A dissociation curve analysis and agarose gel electrophoresis were used to demonstrate that the DNA fragments were amplified to the proper size (S2). A standard curve was created using plasmid DNA with an established copy number for AOB, AOA, or AOB. The PCR efficiency for all tests ranged from 90 to 100%, and the r^2^ value was 0.95 to 99.

### 2.6. Measurement of Dry Matter Accumulation and Grain Yield

To measure dry matter (DM) accumulation, four rows on both sides of the borders were taken at harvesting and used to calculate the amount of DM. The fresh weight was noted, and the samples were kept dry in an oven at 70 °C for 48 h. The samples were then weighed using a digital lab scale. The three undisturbed central rows from each plot were harvested for measurements of the grain yield of mature rice plants; the plants were then dried and weighed, and the yield was calculated. At 14% moisture content, the grain yield was expressed in kg ha^−1^.

### 2.7. Statistical Analysis

The effects of different treatments (low- and high-dose N and their interaction with different biochar rates) on soil chemical properties, enzyme activities, and the abundance of AOA, AOB and NOB were analyzed using Statistics 8.1 software. The significance of the main differences was verified using least significance difference tests at the *p <* 0.05 level. Sigmaplot 12.0 and MS Excel were used to create graphs and tables, respectively. R (3.2) software was run to conduct a correlation analysis of soil properties, enzyme activities, abundances of AOA, AOB, and NOB, plant N concentration, DM production and grain yield.

## 3. Results

### 3.1. Soil Chemical Properties

The addition of biochar under low-dose (135 kg N ha^−1^) and high-dose (180 kg N ha^−1^) urea applications and its effect on soil chemical properties in paddy soil are shown in Table 2. pH, SOC, TN, and mineral nitrogen (NH_4_^+^-N and NO_3_^–^-N) were higher in biochar-amended soil than in unamended soil. The pH in T2 was significantly higher, by 8.63% and 13.73%, than T1 and T5 in the early season, while the pH was 14.38% and 16.37% higher in T4 than in T1 and T5 in the late season, respectively (*p <* 0.05). In the early season, there were no significant variations in the soil water percentage across treatments; however, in the late season, the biochar-applied treatment (T7) had a significantly greater soil water percentage (32.63%) as compared to T1, T2 and T5. The SOC content was highest in both seasons in T4, T7 and T8 among all treatments; specifically, in T4, the SOC content was 25.06% and 45.14% higher than in T1, while in T8, the SOC content was 30.02% and 48.86% higher than in T5 in the early and late seasons, respectively. However, no significant difference was recorded between T1 and T5 in both seasons.

The total N content was higher in biochar-amended soil under both low- and high-dose N applications than in soil not amended with biochar. The total N content was an average of 21.76% and 19.01% higher in both seasons (*p <* 0.05) in T8 than in T1 and T5, respectively (Table 2). The total N content was also found to be higher in T3, T4, and T6; however, no significant differences were observed among them in both seasons (*p >* 0.05). The concentration of NH4^+^-N was higher in both early and late seasons in biochar-amended soil T2, T3, and T4 (at an average of 27.63 mg NH4^+^-N kg^−1^) than in T6, T7, and T8 (at an average of 27.92 mg NH4^+^-N kg^−1^); the concentration of NH4^+^-N was lowest in the solely urea-applied treatments T1 (at an average of 25.27 mg NH4^+^-N kg^−1^) and T5 (at an average of 25.25 mg NH_4_^+^-N kg^−1^). Moreover, a significant difference was observed in the NH_4_^+^-N between T5 and T8 in the early season, although no significant difference was found among treatments with and without biochar application in the late season. The results showed that the NO_3_^–^-N concentration was 17.72% and 18.16% higher in T4 and T8 than in T1 and T5 in the early season, respectively, and these differences were significant. In the late season, treatment T7 documented (11.73 mg kg^−1^) concentration of NO_3_^–^-N, which was significantly higher, by 23.73% and 18.34%, than T1 and T2, respectively. However, no significant variations in NO_3_^–^-N concentration were recorded between T3, T4, T5, T6, T7, and T8.

### 3.2. Soil Enzymes Activities

The activity of soil enzymes, such as dehydrogenase and chitinase, was found higher in the low dose urea application with biochar, but the effect was diminished with the higher dose urea application; the exception was catalase activity, which was found to be non-significant across the treatments in both seasons. (Figure 2B). In the early season, the urease activity was significantly higher by 35.79% in T3 than T1, although T3 was statistically on par (*p* > 0.05) with all treatments except T1. Conversely, in the late season, no significant differences were recorded across the treatments. (Figure 2A). The activity of catalase enzyme was found higher in the early season than in the late season in biochar-added treatments, but lower in both seasons when sole urea was used; however, no significant differences in catalase activity were observed across treatments in either season. Soils amended with biochar had higher polyphenol oxidase activity in the early and late seasons than soils not amended with biochar (*p <* 0.05). The higher polyphenol oxidase activity was recorded in treatment T7 which was 34.72% greater than T5 in the early and 52.04% greater than T6 in late seasons. However, in the early season, no significant differences were found in soil polyphenol oxidase activity among treatments T1, T2, T3, T4, T5, and T6, while in the late season in treatments T1, T2, T3, T4, T5, T6 and T8 (Figure 3A). Soil dehydrogenase enzyme activity increased with the rate of biochar application under low-dose urea (135 kg N ha^−1^); however, the activity of soil dehydrogenase was lowest when biochar was treated with high-dose urea (180 kg N ha^−1^). In the early season, the maximum activity rate was recorded in T4, which was 127.21% and 41.13% higher compared with T1 and T2, respectively, and this difference was significant (*p* < 0.05); however, no significant differences were observed among T1, T5, T6, T7, and T8 (Figure 3B). In the late season, T2 had 138.54% higher dehydrogenase activity than T1; however, with the exception of T2, all treatments were found to be statistically non-significant with T1. Similarly, chitinase enzyme activity was highest in soil amended with biochar under low-dose urea application in both seasons. Moreover, in the early season, the activity of the chitinase enzyme was 39.34% higher in T2 than in T1, and the differences between T2 and the other treatments were significant. In the late season, chitinase activity was 47.54%, 53.54% and 35.94% higher in T3 than in T1, T5, and T7, respectively (Figure 3C). T3 was statistically on par (*p <* 0.05) with all treatments except T1, T5, and T7.

### 3.3. Abundance of AOB

The abundance of AOB in the treatments varied from 1.36 × 10^5^ to 2.33 × 10^5^ copies g^−1^ soil in the early season, and from 1.54 × 10^5^ to 2.31 × 10^5^ copies g^−1^ soil in the late season (Figure 4). Application of urea with biochar significantly altered the abundance of AOB compared with non-applied soil. In the early season, the highest concentration of AOB (2.33 × 10^5^) was found in T4, followed by T6 (2.29 × 10^5^), and the lowest concentrations (1.36 × 10^5^ and 1.52 × 10^5^, respectively) were found in T1 and T2. The abundance of AOB varied significantly between T1, T2, T4 and T6 (*p* < 0.05), although no significant difference was recorded between T3, T4, T5, T6, T7, and T8. AOB was more prevalent in treatments that had biochar added in the late season compared to treatments without biochar. Under the application of low-dose urea (135 kg ha^−1^), there was no significant difference in the abundance of AOB between any of the biochar-applied treatments. However, there was a significant difference in AOB abundance between the biochar-applied treatment T7 (2.31 × 10^5^) and the non-applied treatment T1 (1.54 × 10^5^).

### 3.4. Abundance of AOA

The abundance of AOA ranged from 0.167 × 10^5^ to 0.206 × 10^5^ copies g^−1^ soil in the early season and from 0.141 × 10^5^ to 0.206 × 10^5^ copies g^−1^ soil in the late season (Figure 5). During both seasons, sole urea application or the interaction of urea and biochar application had no significant effect on AOA abundance. The combined application of urea and biochar increased the AOA abundance slightly more than urea application alone. The concentration of AOA was highest (0.206 × 10^5^ copies g^−1^ soil) in T6 in the early season, and it was 15.26% higher than in T5. In the late season, the abundance of AOA was found to be at a minimum (0.141 × 10^5^ copies g^−1^ soil) in T4 and a maximum in T7 (0.206 × 10^5^ copies g^−1^ soil). The abundance of AOA varied among treatments and generally increased with high-dose urea (180 kg N ha^−1^) under biochar application. No significant differences were observed in the abundance of AOA between T7 and the other treatments. There was no discernible trend, and the fertilizer regime had a smaller impact on the abundance of AOA.

### 3.5. Abundance of NOB

The abundance of NOB varied slightly between treatments that applied biochar and those that did not in both seasons, but there were no significant variances in the abundance of NOB between treatments (Figure 6). In the early season, the abundance of NOB was highest (1.53 × 10^5^) in T8, and this increment was 10.72% higher in T8 than T5, (*p >* 0.05). In the late season, the abundance of NOB ranged from 1.1 × 10^5^ copies g^−1^ in T1 to 1.54 × 10^5^ copies g^−1^ soil in T8. Treatments with biochar demonstrated higher soil NOB abundance compared to treatments without biochar. However, there were no significant differences in the soil NOB concentrations under the fertilizer regime.

### 3.6. N Concentration in Stems, Leaves, and Panicles

Table 3 shows variation in the N accumulation in the stems, leaves, and panicles of rice plants in treatments with low- and high-level N application and various rates of application of biochar. Between treatments, there were noticeable changes in N accumulation during both seasons. In the early season, T2 (4.91 g kg^−1^) and T3 (4.88 g kg^−1^) had the highest concentration of N in stems, which was higher by 29.46% and 28.75% than T1, respectively; meanwhile, no significant differences were observed in the N concentration in stems among T2, T3, T4, T6, T7, and T8. Furthermore, the N concentration in the stem in the late season was highest in T4 and T6 (6.16 g kg^−1^), significantly higher, by 32.97% and 13.26%, than T1 and T5, respectively. The N concentration in leaves increased with the rate of biochar application in both seasons, and the highest N concentration was observed in T3 (15.47 g kg^−1^) in the early season and in T4 in the late season (15.64 g kg^−1^). No significant differences were observed in the N concentration in the leaves among treatments (T3, T4, and T7) in the early season and treatments (T3, T4, T6, T7, and T8) in the late season. The N concentration in the panicles was highest in T4 and T8 (11.77 g kg^−1^ and 11.97 g kg^−1^, respectively) in the early season. In the late season, the N concentration in the panicles was highest in T4 (11.90 g kg^−1^) and in T7 (12 g kg^−1^); no significant differences were observed in the N concentration in the panicles among T3, T4, T6, T7, and T8 in both seasons. The N concentration in the panicles was 24.15% and 21.52% higher in T4 and T8 than T1 and T5 in the early season, while in late season, the N concentration in T4 and T7 was 23.82% and 19.63% higher than T1 and T5, respectively; these differences were significant. T1 was found to be statistically non-significant with T5 in both the early and late seasons (*p >* 0.05).

### 3.7. DM Production and Grain Yield

The production of dry matter and grain yield increased intensely when biochar and urea were applied together (Table 4). In the early and late seasons, no discernible variations in grain yield between T3, T4, T7, and T8 were found. DM accumulation was higher in T7 (12,640.67 kg ha^−1^) and T8 (12,834.67 kg ha^−1^), and these differences were statistically significant compared to T1, T2 and T3 in the early season. In the late season, DM accumulation was found to be significantly lower in T1 and T5 (10,018.21 kg ha^−1^ and 11,041.13 kg ha^−1^) than in T7 (12,208.67 kg ha^−1^) (*p* < 0.05). Moreover, T8 had the highest grain yield (8447 kg ha^−1^) in the early growing season, which was 10.02% higher than that of T5. The highest grain production 8365. 95 kg ha^−1^, 8510.40 kg ha^−1^ and 8533.2 kg ha^−1^ during the late season was recorded in treatments T4, T7 and T8, respectively and it was significantly higher than that of T1, T2, T5 and T6. Between T3, T4, T7, and T8, no appreciable changes in maximum grain yield were found. Grain yield and dry matter production were lowest in treatments with no biochar application and biochar applied at the rate of 10 t ha^−1^.

### 3.8. Correlation Heat Map Analysis

A consistent pattern was observed in the relationships between soil properties, enzymatic activities, abundance of AOB, DM, and grain yield (Figure 7). Soil organic carbon was significantly positively related to soil pH (R^2^ = 0.82), total N (R^2^ = 0.77), NO_3_ (R^2^ = 0.62), urease (R^2^ = 0.57), AOB (R^2^ = 0.61), nitrogen uptake (R^2^ = 0.95), dry matter (R^2^ = 0.94), and grain yield (R^2^ = 0.80). Furthermore, there was a positive correlation between the abundance of NO_3_, urease, AOB and grain yield (R^2^ = 0.69, 0.54, 0.63, respectively). There was a negative relationship between catalase activity and total nitrogen, nitrogen uptake, dry matter and grain yield (R^2^ = −0.43, −0.41, −0.44, −0.57), respectively. Heat map analysis showed that soil pH, SOC, urease enzymes and AOB abundance were the key factors affecting the plant nitrogen uptake, dry matter production, and grain yield of rice.

## 4. Discussion

### 4.1. Effect of Urea and Biochar Application on Soil Chemical Properties

The rapidly growing human population and demand for food have resulted in an increase in the use of synthetic N fertilizers, which reduces N use efficiency and causes various environmental problems [30,31]. To mitigate the deleterious effects of chemical N fertilizers on soil quality and health, we studied the physical, chemical, and biological properties of soil and plant N uptake in a rice paddy field in the early and late seasons in 2020 under low- and high-dose urea application and different levels of biochar application. Soil pH and concentrations of SOC, total nitrogen, and nitrate were higher in 20 and 30 t B ha^−1^, applied with 135 or 180 kg N ha^−1^, than in treatments applied solely with urea (T1 and T5). One possible explanation for the higher soil pH under combined application of biochar and urea might be the alkaline nature of biochar; the porous structure of biochar and its high inner surface area might be responsible for the observed enhancement of soil chemical properties [32]. Moreover, the probable reason for the improvement of the soil pH and SOC in the current study is due to the biochar used in the experimental investigation having high porosity, the presence of hydrophilic domains, and a significant specific surface area; all of these factors have an impact on soil properties. Biochar application can decrease fertilizer requirements and increase microbial activity, soil water-holding capacity, and SOC content, which can enhance the physical properties of soil [33,34].

Biochar application improves SOC and causes significant increases in the TN, and NO_3_^–^-N as compared to NH_4_^+^-N (Table 2). This outcome could be attributed to the experimental site’s limiting of NH_4_^+^ which serves as the substrate for microorganisms that oxidize ammonia, as well as biochar’s capacity to promote nitrification through the adsorption of nitrification-inhibiting chemicals including phenols and terpenes. According to Biederman and Harpole [35], the improvements in soil nutrients are associated with increases in the SOC content mediated by biochar application, as well as mineral substances dissolved in soil solutions. Biochar has been reported to increase soil pH and promote the retention of plant-available water because of its high specific surface area and alkalinity [36,37]. We observed significant differences in water concentration, carbon and nitrogen stocks among treatments varying in levels of urea and biochar application (*p <* 0.05). This indicates that biochar elevates soil organic carbon levels, resulting in high carbon to nitrogen ratios, which could promote soil nitrification and enhance nitrogen’s bioavailability. Rashid et al. [37] discovered that biochar can improve soil C reserves, encourage the retention of soil nutrients, and maximize soil fertility, which is in line with our findings. Utilizing biochar increases the effectiveness of utilizing both nutrients and water [38]. According to Lima et al. [39], using biochar improves the efficiency of using both nutrients and water.

### 4.2. Effect of Urea and Biochar Application on Soil Enzyme Activities

The capacity for biochemical reactions, soil microbial activity, nutrient cycling, and material metabolism can all be measured using the soil enzyme activity as a reference index. Due to their great sensitivity to environmental changes, soil enzymes’ activity can reveal changes in soil quality in a variety of situations [40]. In our study, some of the tested soil enzyme (urease, polyphenol and dehydrogenase) activities were higher in biochar-amended soils than in soils not amended with biochar; the only exception was catalase activity in the early and late seasons. This could be as a result of the close relationship between soil enzyme activity and soil nutrients’ richness, microbial diversity, and organic matter content. According to previous studies, biochar addition might stimulate soil enzyme activity by (1) altering the physicochemical properties of soil [41], (2) absorbing substrates on its surface, or (3) inhibiting the enzymes’ reaction sites [9,42]. In the current study, in the early season, a positive effect was observed between biochar and N fertilizer on soil urease, polyphenol oxidase at 20 t B ha^−1^ and 180 kg N ha^−1^ and dehydrogenase activity at 30 t B ha^−1^ and 135 kg N ha^−1^. A key regulator of the nitrogen cycle in soil is urease; it is primarily involved in stimulating the hydrolysis of urea and acts as a fundamental reference signal for determining the quantity and ability of soil nitrogen mineralization. Urease activity was highest in biochar-applied soil. Given that the application of biochar promotes N mineralization, N-cycling enzyme activity increases with biochar application [42], as was observed in the positive correlation of urease with SOC in our study (R^2^ = 0.57) (Figure 7). Soil enzyme activities might be affected by soil pH, which aligns with the results of an earlier study [43]. Zhao et al. [44] reported that root growth stimulated by biochar can lead to the excretion of enzymes into the soil.

In our study, soil polyphenol oxidase and dehydrogenase was higher with biochar applied at 20 and 30 t ha^−1^, compared with only urea at 135 kg ha^−1^ and 180 kg ha^−1^. The activities of these enzymes increased with the rate of biochar application, and a similar pattern was observed in SOC. This could possibly be the result of increases in dissolved and soil organic matter following the application of biochar, which improves the diversity and number of soil microbes and the activity of enzymes [23]. Therefore, we speculate that the activities of enzymes increased due to increases in SOC induced by biochar application. Similar results for C-cyclase enzymes, which are present on the surface of biochar and are implicated in the colocalization and stability of C compounds, were published by Foster et al. [45]. Oxidoreductase enzymes include polyphenol oxidase and dehydrogenase. They play a significant role in the cycling of soil aromatic compounds and are mostly obtained from plant wastes, root exudates, and soil microbes [46]. Both soil organic carbon and active organic carbon, which are important markers that influence soil enzyme activity, are increased by biochar. Soil polyphenol oxidase and dehydrogenase have different activities, and application of biochar amplified each of these enzyme activities, which is similar to the findings of Huang et al. [47]. The augmentation of carbon sources and nutrient content which may have been immobilized by soil microbes must have been one of the primary causes of this increase (Table 2). As a result, the addition of a significant amount of biochar and urea increased soil nitrogen concentration and entertained a microbial niche. In contrast to these findings, biochar had no more impact on soil catalase activity in both early and late seasons than it did on other N and C-cycling enzymes; this could be attributed to seasonal variation as well as the physical and chemical composition of the soil [23]. The results of our study indicate that biochar application plays a key role in enhancing the activities of soil enzymes in paddy soil under high- and low-dose N fertilizer applications.

### 4.3. Effect of Urea and Biochar Application on the Abundances of AOB, AOB, and NOB

The abundance of AOB increased significantly with the rate of biochar application under low-dose urea application; it was also high under all levels of biochar application and high-dose urea application (*p <* 0.05), and this was associated with the higher NO_3_^–^-N concentration in these soils (Table 2). The abundances of AOA and NOB were slightly higher in the N fertilizer + biochar treatments than in the N fertilizer treatments alone; however, no significant differences in AOA and NOB abundances were observed between these two sets of treatments (Figure 5 and Figure 6). These results imply that by increasing the abundance of AOB in paddy soil with combined application of N and B fertilizer, one can enhance nitrification. According to the results of a previous study conducted in natural and alkaline soil, AOB plays a greater role in soil nitrification than AOA when N fertilizer is applied [48,49]. However, the effects of biochar on the abundances of AOA and AOB differed among treatments in both seasons. For example, there was significant variation among treatments in the abundance of AOB, and biochar addition had no significant effect on the abundance of AOA. Unlike our findings, Prommer et al. [50] documented that biochar addition increased the abundances of both AOB and AOA in agricultural soil, which enhanced soil potential nitrification rates. Similarly, Song et al. [51] found that higher abundances of AOA increased the soil ammoxidation rate in coastal saline soil. Increases in AOB and AOA abundances due to biochar application could have several explanations.

First, biochar has a vast surface area and a highly porous structure which, along with its strong ability to hold water and retain nutrients, facilitate the provision of resources that meet the particular metabolic requirements of microbes [52]. Second, biochar can enhance living conditions for soil microorganisms by increasing the pH of soil [53]. Third, the SOC and NO_3_ in biochar enhance soil AOB abundance (R^2^ = 0.61 and R^2^ = 0.47, respectively; Figure 7). Finally, biochar might absorb substances that inhibit nitrification, such as polyphenols or tannins [53,54]. However, a few studies have shown that biochar application has no effect or even a negative effect on soil nitrification [55,56]. This could be due to the release of nitrification inhibitors such as ethylene and pinene, which reduce the activity of soil AOA and AOB [54,57], and the effect varies depending on the parent materials and biochar formation processes used. In our study, the abundance of NOB varied among the biochar-applied treatments; however, differences in the abundance of NOB among treatments were not significant, indicating that AOB was more sensitive to biochar application in paddy soil compared with AOA and NOB. Much research consistent with our findings, has shown that the addition of N fertilizer increases the abundance of AOB, but not AOA or NOB [58,59].

### 4.4. Effect of Urea and Biochar Application on Plant N Uptake, DM, and Grain Yield

Nitrogen is usually the most yield-limiting nutrient in rice production, and its uptake by rice is closely associated with N presence and loss in soil [60]. Nitrogen accumulation in biochar-applied treatments was significantly higher in biochars 20 and 30 t ha^−1^, applied with 135 kg N ha^−1^ and 180 kg N ha^−1^, in the early and late seasons, than treatments applied solely with urea (135 kg N ha^−1^). The increase in N uptake might stem from improvements in the soil’s physiochemical properties and increases in microbial activity due to the high rate of biochar application [61]. Furthermore, no significant differences were found in the leaf and panicles nitrogen concentration between low-dose and high-dose N fertilizer applications without biochar. Our findings are consistent with those of Ali et al. [62], showing that N accumulation increases under high rates of biochar application along with N fertilizer application. Similarly, Huang et al. [60] revealed that biochar application to paddy soil enhances the uptake of fertilizer N (23–27%). Biochar application affects root morphological characteristics, enhances DM accumulation, and increases plant N uptake [33]. Huang et al. [63] showed that the application of biochar to paddy fields increases N uptake in rice and improves N use efficiency. The accumulation of plant-available nutrients in biochar can lead to increases in the total N content in soil [60].

In our study, dry matter production was significantly higher in biochar-applied treatments than in treatments in which biochar was not applied in the early season. Although, in late season maximum dry matter produced in treatments applied with 30 t B ha^−1^ with low dose urea (T4) and biochar 20 and 30 t ha^−1^ with high dose urea applied (T7 and T8). Increases in DM can be attributed to improvements in soil properties (Figure 7, SOC = R^2^ = 0.94, NO_3_ = 0.63) and enzymatic activities due to biochar application (Table 2), which enhances plant N uptake and DM production (Table 3 and Table 4). Possible explanations for these increases include the positive effects of biochar and N fertilizer on soil fertility and DM production in rice [33,34], which enhance plant growth [64]. Another possible reason is that biochar improves plant DM and N uptake [61,65]. Zahoor et al. [66] reported that increases in the supply of N fertilizer can increase biomass enzymatic saccharification and DM production in rice. The yield of cereal crops depends on plant growth and N accumulation; a previous study has shown that DM accumulation is responsible for 23–68% of grain yield, which is in turn determined by soil properties and agronomic practices [67]. It is likely that in the current study, rice grain yield was highest in the combined application of biochar 20 and 30 t ha^−1^ and urea 135 and 180 kg ha^−1^ in the early and late seasons, respectively. Rice grain yield was found to be lowest in sole urea application (135 kg N ha^−1^) in both seasons, and it varied significantly across treatments. The increase in rice grain yield under biochar application might have fluctuated with enhanced soil physiochemical properties, soil microbial biomass, and the rate of photosynthesis under biochar addition. Although an increase in soil pH due to the addition of biochar generally increases the richness of AOB and alkaline N in acidic soil [68], another study has shown that application of biochar formed from coconut husk enhances nutrient retention by decreasing the abundance of nitrifiers and the nitrification process, which ultimately increases grain yield [64,69]. Biochar addition increased N uptake, which resulted in a positive interaction with DM production and grain yield (Figure 7). Our findings are in line with the results of Ali et al. [33], showing that nitrogen accumulation and grain yield are highest under high rates of biochar application applied with N fertilizer in paddy rice.

## 5. Conclusions

Our results validate that the combined application of urea fertilizer and biochar enhanced soil chemical properties, enzymes activity, and the abundance of AOB. Ultimately, it also improved N uptake by rice plants, DM production, and grain yield in both early and late seasons. The findings show that there are no significant (*p <* 0.05) differences between treatments utilizing urea at low (135 kg ha^−1^) and high (180 kg ha^−1^) doses in combination with 20 or 30 t ha^−1^ of biochar in terms of soil NO_3_ concentrations, plant N uptake, or grain production. Therefore, in double-cropping rice systems, urea application at 135 kg ha^−1^ with biochar 20 or 30 t ha^−1^ provides appropriate soil C and N stocks, improves the microbial community, and raises rice yield. Additional research is required to determine how biochar application could be transformed over time in paddy soil to increase rice yield.

## Figures and Tables

**Figure 1 microorganisms-11-00527-f001:**
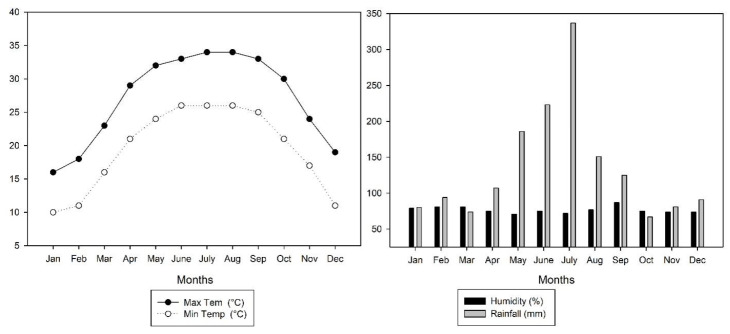
Mean maximum and minimum temperature, relative humidity, and total rainfall during the early and late seasons in 2020.

**Figure 2 microorganisms-11-00527-f002:**
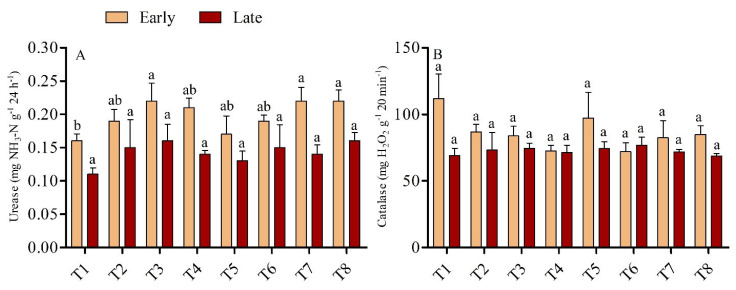
(**A**,**B**): Soil urease (**A**) and catalase (**B**) activity during the early and late seasons under low- and high-dose urea application and different levels of biochar application. Note: T1: N1B0 = 135 kg N ha^−1^ + 0 t B ha^−1^; T2: N1B1 = 135 kg N ha^−1^ + 10 t B ha^−1^; T3: N1B2 = 135 kg N ha^−1^ + 20 t B ha^−1^; T4: N1B3 = 135 kg N ha^−1^ + 30 t B ha^−1^; T5: N2B0 = 180 kg N ha^−1^ + 0 t B ha^−1^; T6: N2B1 = 180 kg N ha^−1^ + 10 t B ha^−1^; T7: N2B2 = 180 kg N ha^−1^ + 20 t B ha^−1^; and T8: N2B3 = 180 kg N ha^−1^ + 30 t B ha^−1^. Bars with different letters are significantly different, at *p <* 0.05.

**Figure 3 microorganisms-11-00527-f003:**
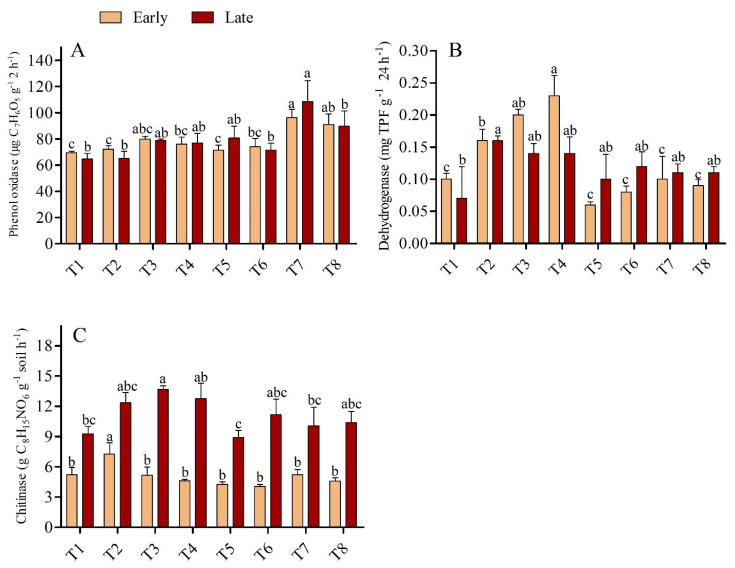
(**A**–**C**): Soil polyphenol oxidase enzyme (**A**), dehydrogenase enzyme (**B**), and chitinase enzyme (**C**) activity during the early and late seasons under low- and high-dose urea application and different levels of biochar application. Note: Bars with different letters are significantly different at *p <* 0.05.

**Figure 4 microorganisms-11-00527-f004:**
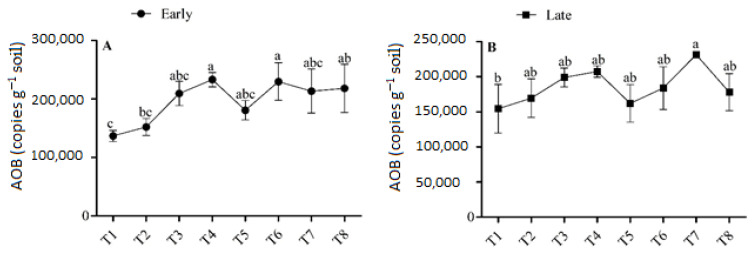
(**A**,**B**) Concentration of AOB in the early and late season under high- and low-dose urea application and different levels of biochar application. Note: Bars with different letters are significantly different, at *p <* 0.05.

**Figure 5 microorganisms-11-00527-f005:**
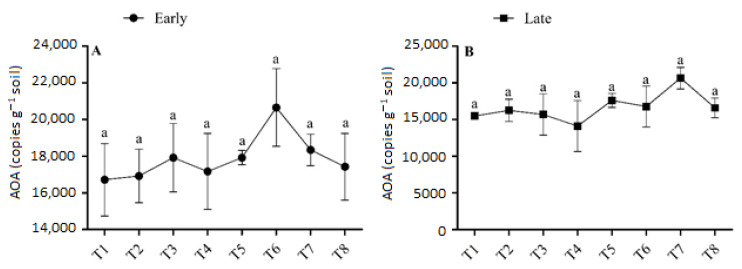
(**A**,**B**) Concentration of AOA in the early and late season under high- and low-dose urea application and different levels of biochar application. Note: Bars with different letters are significantly different, at *p <* 0.05.

**Figure 6 microorganisms-11-00527-f006:**
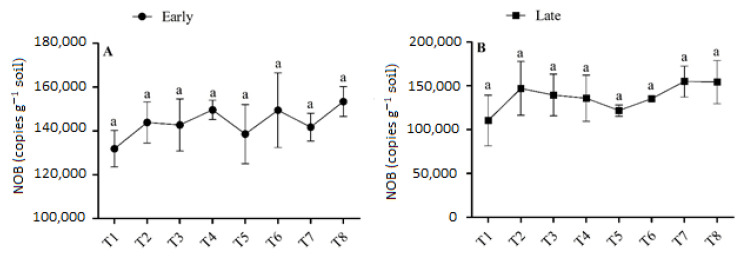
(**A**,**B**) Concentration of NOB in the early and late seasons under low- and high-dose urea application and different levels of biochar application. Note: Bars with different letters are significantly different, at *p <* 0.05.

**Figure 7 microorganisms-11-00527-f007:**
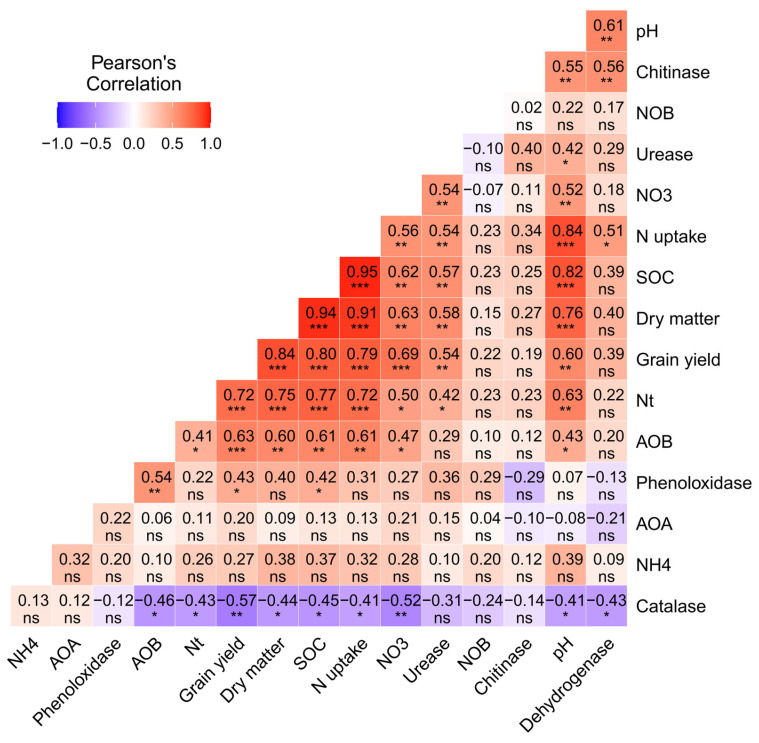
Correlation heat map of soil properties; enzyme activities; abundances of AOB, AOA, and NOB; N uptake; DM; and grain yield of rice. Red colors indicate positive relationships, and dark blue colors indicate negative relationships. The colors carried from white to red and white to blue indicate moderately to strong positive/negative correlations, respectively. Note: SOC; soil organic carbon, Nt; total nitrogen, NH_4_; ammonium nitrogen concentration, NO_3_; nitrite nitrogen concentration, AOB; ammonium-oxidizing bacteria, AOA; ammonium-oxidizing archaea, NOB; nitrite-oxidizing bacteria. * indicate the significant difference at (*p <* 0.05), ** indicate the significant difference at (*p <* 0.01), and *** indicate the significant difference at (*p <* 0.001), whereas ns indicate non-significant (*p ≥* 0.05).

**Table 1 microorganisms-11-00527-t001:** Rate and timing of biochar and urea application.

Treatment	Urea (kg ha^−1^)	Biochar (t ha^−1^)	Biochar (kg plot^−1^)	Urea Basal (g plot^−1^)	Urea Tillering (g plot ^−1^)	Urea Panicle Initiation (g plot^−1^)
(T1) N1B0	135	0	0	343.36	206.02	137.34
(T2) N1B1	135	10	23.4	343.36	206.02	137.34
(T3) N1B2	135	20	46.8	343.36	206.02	137.34
(T4) N1B3	135	30	70.2	343.36	206.02	137.34
(T5) N2B0	180	0	0	457.82	274.69	183.12
(T6) N2B1	180	10	23.4	457.82	274.69	183.12
(T7) N2B2	180	20	46.8	457.82	274.69	183.12
(T8) N2B3	180	30	70.2	457.82	274.69	183.12

**Table 2 microorganisms-11-00527-t002:** Responses of soil chemical properties to low- and high-dose N application with and without biochar application.

Treatments	pH	Water %	SOC (g kg^−1^)	TN (g kg^−1^)	NH4^+^-N (mg kg^−1^)	NO3^–^-N (mg kg^−1^)
Early season						
T1	6.02 ± 0.04 d	32.17 ± 2.34 a	9.90 ± 0.07 e	1.19 ± 0.04 e	25.95 ± 0.89 ab	8.42 ± 0.12 d
T2	6.54 ± 0.06 a	31.01 ± 1.67 a	10.98 ± 0.05 d	1.29 ± 0.05 cde	29.63 ± 1.05 ab	9.80 ± 0.65 abc
T3	6.45 ± 0.06 ab	31.99 ± 3.12 a	11.91 ± 0.05 c	1.34 ± 0.06 cd	28.22 ± 2.56 ab	9.94 ± 0.55 abc
T4	6.48 ± 0.03 ab	30.46 ± 2.27 a	12.39 ± 0.03 b	1.38 ± 0.04 bcd	27.18 ± 1.52 ab	9.92 ± 0.57 abc
T5	5.75 ± 0.04 e	31.54 ± 1.89 a	9.96 ± 0.04 e	1.25 ± 0.05 de	25.33 ± 0.78 b	8.81 ± 0.27 cd
T6	6.26 ± 0.06 c	29.72 ± 2.78 a	11.91 ± 0.05 c	1.43 ± 0.05 abc	25.97 ± 1.38 ab	9.11 ± 0.23 bcd
T7	6.34 ± 0.06 bc	29.58 ± 3.12 a	12.61 ± 0.05 ab	1.49 ± 0.05 ab	29.94 ± 3.21 ab	10.24 ± 0.17 ab
T8	6.41 ± 0.02 abc	31.05 ± 2.12 a	12.95 ± 0.06 a	1.53 ± 0.04 a	32.30 ± 2.32 a	10.41 ± 0.16 a
Late season						
T1	5.84 ± 0.06 c	30.27 ± 2.89 b	9.57 ± 0.03 f	1.22 ± 0.04 c	24.60 ± 1.95 a	9.48 ± 0.08 b
T2	6.26 ± 0.07 b	30.96 ± 3.72 b	12.55 ± 0.05 e	1.39 ± 0.04 ab	28.07 ± 3.21 a	9.91 ± 0.25 b
T3	6.35 ± 0.06 b	31.27 ± 1.98 ab	13.41 ± 0.07 d	1.41 ± 0.5 ab	26.41 ± 1.25 a	10.26 ± 0.64 ab
T4	6.68 ± 0.06 a	31.39 2.89 ab	13.89 ± 0.04 bc	1.43 ± 0.05 a	26.28 ± 1.06 a	10.46 ± 0.52 ab
T5	5.74 ± 0.06 c	30.58 ± 2.67 b	9.71 ± 0.11 f	1.28 ± 0.04 bc	25.16 ± 2.32 a	9.97 ± 0.77 ab
T6	6.26 ± 0.05 b	31.58 ± 3.12 ab	13.81 ± 0.07 c	1.38 ± 0.04 ab	26.28 ± 1.54 a	10.91 ± 1.08 ab
T7	6.29 ± 0.06 b	32.63 ± 2.18 a	14.11 ± 0.06 b	1.46 ± 0.04 a	29.96 ± 1.78 a	11.73 ± 0.42 a
T8	6.34 ± 0.07 b	31.21 ± 1.89 ab	14.46 ± 0.08a	1.48 ± 0.02 a	26.06 ± 1.48 a	10.23 ± 0.38 ab

Note: Treat-treatment; ES-early season; LS-late season; T1: N1B0. = 135 kg N ha^−1^ + 0 t B ha^−1^; T2: N1B1 = 135 kg N ha^−1^ + 10 t B ha^−1^; T3: N1B2 = 135 kg N ha^−1^ + 20 t B ha^−1^; T4: N1B3 = 135 kg N ha^−1^ + 30 t B ha^−1^; T5: N2B0 = 180 kg N ha^−1^ + 0 t B ha^−1^; T6: N2B1 = 180 kg N ha^−1^ + 10 t B ha^−1^; T7: N2B2 = 180 kg N ha^−1^ + 20 t B ha^−1^; and T8: N2B3 = 180 kg N ha^−1^ + 30 t B ha^−1^; pH-potential hydrogen; SOC-soil organic carbon; TN-total nitrogen; NH_4_^+^-N-soil ammonium nitrogen; NO_3_^–^-N-soil nitrate-nitrogen. Values followed by the different letters within columns are significantly different, at *p <* 0.05.

**Table 3 microorganisms-11-00527-t003:** Response of plant N accumulation to low- and high-dose urea application under different levels of biochar application.

Treatments	Stem g N kg^−1^	Leaves g N kg^−1^	Panicles g N kg^−1^
Early season			
T1	3.79 ± 0.06 c	11.90 ± 0.34 d	9.48 ± 0.36 c
T2	4.91 ± 0.17 a	13.82 ± 0.17 c	10.37 ± 0.09 b
T3	4.88 ± 0.12 a	15.47 ± 0.29 a	11.60 ± 0.32 a
T4	4.87 ± 0.17 a	14.83 ± 0.20 ab	11.77 ± 0.27 a
T5	3.92 ± 0.09 bc	12.41 ± 0.27 d	9.85 ± 0.30 bc
T6	4.76 ± 0.21 a	14.38 ± 0.07 bc	11.72 ± 0.31 a
T7	4.43 ± 0.19 ab	14.80 0.17 ab	11.78 ± 0.32 a
T8	4.47 ± 0.30 a	14.32 ± 0.32 bc	11.97 ± 0.20 a
Late season			
T1	4.64 ± 0.20 c	12.40 ± 0.28 c	9.61 ± 0.36 c
T2	6.09 ± 0.08 a	14.53 ± 0.30 b	10.50 ± 0.09 b
T3	6.01 ± 0.06 a	15.44 ± 0.21 a	11.73 ± 0.32 a
T4	6.17 ± 0.17 a	15.64 ± 0.12 a	11.90 ± 0.27 a
T5	5.05 ± 0.09 b	12.81 ± 0.27 c	10.03 ± 0.25 bc
T6	6.17 ± 0.11 a	15.25 ± 0.11 a	11.96 ± 0.37 a
T7	6.03 ± 0.05 a	15.20 ± 0.17 ab	12.00 ± 0.23 a
T8	6.16 ± 0.10 a	15.31 ± 0.26 a	11.90 ± 0.06 a

Note: Values followed by different letters with in the columns are significantly different at *p <* 0.05.

**Table 4 microorganisms-11-00527-t004:** Dry matter production and grain yield under low- and high-dose urea application and different rates of biochar application.

Treatments	Dry Matter kg h^−1^	Grain Yield kg h^−1^
Early season		
T1	10,764.72 ± 177.14 c	6621.32 ± 144.21 c
T2	12,340.67 ± 139.90 b	7678.06 ± 164.28 b
T3	12,434.23 ± 160.66 ab	8032.76 ± 43.46 ab
T4	12,400.67 ± 164.39 ab	8418.80 ± 177.92 a
T5	11,067.33 ± 121.21 c	7678.06 ± 239.63 b
T6	12,694.24 ± 55.80 ab	7799.15 ± 140.11 b
T7	12,640.67± 57.95 ab	8425.93 ± 126.01 a
T8	12,834.67 ± 159.68 a	8447.29 ± 135.88 a
Late season		
T1	10,018.21 ± 149.72 d	6605.84 ± 78.01 d
T2	11,560.00 ± 147.35 b	7910.11 ± 164.28 c
T3	11,402.33 ± 152.77 bc	8264.81 ± 97.23 abc
T4	12,072.67 ± 130.67 a	8365.95 ± 124.18 ab
T5	11,041.13 ± 178.63 c	7910.11 ± 139.63 c
T6	11,525.23 ± 113.80 b	8031.20 ± 140.11 bc
T7	12,208.67 ± 88.51 a	8510.40 ± 90.62 a
T8	12,115.67 ± 131.68 a	8533.19 ± 124.41 a

Note: Columns with different letters among treatments are significantly different, at *p <* 0.05.

## Data Availability

The data will be made available on request.

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
