# Peer review of "Partial Substitution of Urea with Biochar Induced Improvements in Soil Enzymes Activity, Ammonia-Nitrite Oxidizers, and Nitrogen Uptake in the Double-Cropping Rice System"

_microorganisms, 2023, doi:10.3390/microorganisms11020527_

Round 1
Reviewer 1 Report
Manuscript by Ullah et al. reports the effects of urea and biochar application on bio-chemical properties of soil; N uptake, and the grain yield of rice. This study appears well-planned and well-executed. Manuscript also falls within the scope of the Journal. However, in order to improve quality of the manuscript, I would like to recommend following changes before it is accepted for publication:
Title: „Partial substitution of biochar with urea“ or conversely?
Abstract: „However, few studies have characterized the effects of urea and biochar application on soil and rice plants in paddy fields“ – soil and plants, in what way?
Abstract: „under low (135 kg ha-1 ) and high (180 kg ha-1 ) levels of urea application and different rates of biochar (0 t ha-1 , 10 t ha-1 , 20 t ha-1 , and 30 t ha-1 ) application“ – include more information about experimental design.
Abstract: The enzymes urease activity, phenol oxidase activity, dehydrogenase activity, and chitinase activity were 25.28%, 14.13%, 67.76%, and 22.26% higher on average, and soil catalase activity was 15.06% lower on average in biochar-amended soil than in soil not amended with biochar – rewrite sentence – e. g. On average, the activity of urease, phenol oxidase, dehydrogenase, and chitinase enzymes were...
Introduction: Rice was cultivated on more than 1.61 × 108 ha of land worldwide in 2015–2016 –include more recent data.
Introduction: In addition B has a crucial part in the soil N cycle by decreasing the leaching of inorganic N and nitrous oxide (N2O) emissions [11], which promotes organic nitrogen fixation [12] – what is „B“? Write full name.
Introduction: „Nitrification one of the crucial process in the N cycle's, involves...“ – is one... and involves..
Introduction: „Few research, meanwhile, have examined how biochar application rates affect AOB, AOA, and NOB abundances“ – Which research? Include citation.
Materials and Methods: Include more information about cultivar.
Materials and Methods: „Seeds were grown in pastic trays“ – plastic trays
Materials and Methods: „The same agronomic procedures were used in all plots throughout the experiments, including irrigation, insecticides, and herbicides“ – include more information about applied agronomic procedures.
Materials and Methods: Soil and plant sampling – authors should include more information about method of soil sampling, number od samples...
Materials and Methods: Include information about repetition for measurement/analyses of biological and chemical traits
Materials and Methods: „At the end of the PCR analysis, melting curve analysis and agarose gel electrophoresis were used to evaluate the product specificity“ – include more information about each procedure.
Materials and Methods: „Effects of different treatments (low- and high-dose N and their interaction with different biochar rates) on soil chemical properties, enzyme activities, and the abundance of ammonia-nitrite oxidizers were analyzed using Statistics 8.1 software“ – and the abundances of AOA, AOB?
Results: „which was sometimes lower in biochar-amended soil in the early season than in the late season“ – not a specific effect, rewrite.
Results: „In the fertilizers regime, there were no appreciable variations in soil NOB concentrations“ - find better words for appreciable/concentrations.
Discussion/Conclusion: Describe treatments instead of using „T“ in these sections.
Discussion: Emphasize the importance of analyzed soil enzymes.
Conclusion: This section should be concise, focusing on two/three ‘concluding’ remarks rather than being a repetition of results.
Supplementary file (S1): Include reference for primers or move this table from supplementary in the main text.
Supplementary file (S1): AOB AOA NOB – explaination for abbreviations is missing bellow the table; also, is it correct to write „strain“ here?
Author Response
We are very thankful that you provided your valuable suggestions and gave us the opportunity to revise our manuscript. We have kindly asked the authors to revise the manuscript. Our point by point response to all the comments and suggestions is given below. Moreover, for better understanding, the changes are marked with track changes in the revised manuscript. We hope that you will find the revised manuscript acceptable for publication in Microorganisms.

Reviewer 2 Report
In this manuscript, authors studied the changes in soil bio-chemical properties, plant nitrogen (N) uptake, and the grain yield of rice in the early and late seasons of 2020 under low (135 kg N ha-1) and high (180 kg N ha-1) levels of urea application with biochar (0 t ha-1, 10 t ha-1, 20 t ha-1, and 30 t ha-1). The results revealed that soil pH, soil organic carbon content, total N content, and mineral N content (NH4+-N and NO3−-N) were improved with biochar. The enzyme activities were stimulated with biochar addition, and also the bacteria and archaea influencing N cycling were increased with the combined application of biochar and urea. The findings offer a novel concept and theoretical foundation for using biochar in double-cropping rice systems in the subtropics to reduce N input and increase N use efficiency by improving biological properties. The aims of the authors are substantial, and the results are properly documented. However, the authors need to address the following concerns before getting it suitable for publication in "Microorganisms".
1. The manuscript needs grammar checking and paraphrasing editing. The English language should be checked by a native English speaker.
2. The title can be improved.
3. The abstract can be shorten according to the scientific merit, with principal objectives and major conclusions.
4. Please, try to improve the introduction by adding your hypothesis and rational objectives at the end. Meanwhile, add more relevant details in the introduction that how biochar improves the soil biological functions with some recent references.
5. The soil of the experimental plot is… Change to experimental site.
6. What are the justifications of choosing (low 135 kg ha-1 and high 180 kg ha-1) levels of nitrogen? References?
7. What are the sources used for phosphorus and potassium fertilizer?
8. The description of the experimental operation can be shorten in sections 2.4.
9. Recheck Table 1, it contains uppercase letters.
10. Seedlings from 15 hills in each plot were randomly collected for measurements of plant N concentrations… Revise the sentence.
11. For measurement of dry matter production the rows taken from which side of the plot, containing how many hills? Please elaborate your connotation.
12. It would be nice if the authors could add plant N uptake in the correlation analysis.
13. In Section 3, authors have resulted water percentage. It is suggested to add the relevant procedure followed for the measurement of water percentage in the methodology section.
14. No significant differences were observed in NH4 +–N among treatments with and without biochar application. Please clarify your sentence about the seasons because from Table 2 it seems a significant difference in the early season.
15. The concentration of NO3 ––N was higher in Table 4 in the early season, recheck the sentence because the concentration is higher in Table 3 followed by Table 4 in the low dose urea application.
16. Section 3.2. Catalase activity was lower in the late season than in the early season when biochar was not added…. It’s the repetition of the above sentences, revise the results of catalase activity.
17. Remove the inside borders from Table 4.
18. The conclusion should be concise and provide a clear idea about the significance of recommended low dose urea application instead of high dose urea.
Author Response
Thank you for your kind suggestion; this helped us a lot in improving the manuscript. The author’s responses are enlisted one by one, and the changes are made in the revised manuscript. Please find the rebuttal in the attached file. We hope that you will find the revised manuscript acceptable for publication.

Reviewer 3 Report
Manuscript ID: microorganisms-2187338
Type of manuscript: Article
Title: Partial substitution of biochar with urea increases enzymes activity, the abundance of ammonia-nitrite oxidizers, and nitrogen uptake in a double-cropping rice system
Authors: Saif Ullah *, Izhar Ali, Mei Yang, Quan Zhao, Anas Iqbal, Xiaoyan Wu, Shakeel Ahmad, Ihsan Muhammad, Abdullah Khan, Muhammad Adnan, Pengli Yuan, Ligeng Jiang
Submitted to section: Microbial Biotechnology,
General and Major Comments
The authors studied effects of the addition of biochar and urea on a lot of parameters such as soil properties, enzymes activities, abundance of nitrification microorganisms, nitrogen content in rice, and the growth of rice. They showed that the addition and amount of biochar and urea affected these parameters. The authors described positive effects of the application of biochar.
Although I understand that the authors examined a lot of parameters and that the addition of biochar significantly affected several parameters in some cases, I am afraid that this manuscript is not acceptable for the following reasons.
1. The interpretation of the results is not objective or scientific. Even in the absence of significant differences, the authors stated that the biochar elevated some parameters. The authors also expanded on the conditions where there were some significant differences and stated that biochar made the parameters higher. Additionally, explanation of results was unclear in some areas. Some of them are as follows. (I can't list them all, but there were similar areas in need of improvement throughout the result, discussion, and conclusion chapters.)
Page 6: pH, SOC, TN, and… amended with biochar.; There was no significant difference in NH4+-N in most cases in Table 2. (There were other similarly inappropriate expressions in the manuscript. For example, Page 15: We found that … and late seasons., Page 16: There was a … AOA and NOB.)
Page 6: No significant in soil … amended with biochar.; The authors mentioned no significant difference, but they described the water percentage was higher. (There were other similarly inappropriate expressions in the manuscript.)
Page 6: the SOC content … late season, respectively.; Even though T8 was compared with T1 and T5 in two seasons, only two percentages were listed. (There were other similarly inappropriate expressions in the manuscript.)
2. In the discussion chapter, although the authors describe the results of many previous papers, they did little to discuss this study. The authors need to discuss the relevance and differences between the previous results and the current results, and what this study has revealed.
Since the authors measured many parameters and in some cases the application of biochar and urea had a significant effect, the authors should reconsider the results and discussions, especially the presence or absence of significant differences, and then resubmit it.
I hope these comments will be helpful.
Author Response
Respected Reviewer,
Thank you for giving us the opportunity to revise our manuscript. Please find the rebuttal of our revised manuscript entitled “Partial substitution of urea with biochar induced improvements in soil enzymes activity, ammonia-nitrite oxidizers, and nitrogen uptake in the double-cropping rice system” in the attached file. We are grateful for the critical review of our manuscript. We found that all comments are very constructive and will more strengthen our manuscript. Therefore, we have incorporated all comments and made all the changes, accordingly. We have thoroughly revised the entire manuscript and addressed all questions.

Round 2
Reviewer 2 Report
I am satisfied with the authors' correction on the ms. Here, I suggest a accept after minor revision on some spelling.
Author Response
Thank you. We have revised the manuscript in accordance with your kind suggestions.
Reviewer 3 Report
Manuscript ID: microorganisms-2187338
Type of manuscript: Article
Title: Partial substitution of biochar with urea increases enzymes activity, the abundance of ammonia-nitrite oxidizers, and nitrogen uptake in a double-cropping rice system
Authors: Saif Ullah *, Izhar Ali, Mei Yang, Quan Zhao, Anas Iqbal, Xiaoyan Wu, Shakeel Ahmad, Ihsan Muhammad, Abdullah Khan, Muhammad Adnan, Pengli Yuan, Ligeng Jiang
Submitted to section: Microbial Biotechnology,
General and Major Comments
Although I understand that the authors revised their manuscript in response to review comments, I am afraid that this manuscript is not acceptable for the following reasons. Perhaps it is because my comments were not detailed or my English was not clear, but the authors did not properly revise in response to my previous comments.
1. The interpretation of the results is still not objective or scientific. Especially, it is not appropriate for a scientific paper to state that the results were higher or lower, even though there is no significant difference. If there was no significant difference, then it cannot be said to “be higher”. If there were conditions with no significant differences, we cannot ignore them and say that “they were higher”. It should be clearly stated under which conditions there was a significant difference.
In the manuscript, even in the absence of significant differences, the authors stated that the biochar made some parameters higher. The authors also expanded on the conditions where there were some significant differences and stated that biochar made the parameters higher. Additionally, explanation of results was unclear in some areas. The presentation of averages or percentages of values for different conditions is either inappropriate or unclear in intent. Examples in page 7 to 8 are listed below.
Page 7: pH, SOC, TN, and … in unamended soil.; There was no significant difference in TN in some cases (T1 and T2 in early season, T5 and T6 in late season), in NH4+-N in most cases, in NO3--N in some cases (T5 and T6 in early season, almost all combinations in late season) in Table 2. It is not appropriate to describe that these parameters were higher in the biochar-amended soil, by focusing only on the areas where there were significant differences.
Page 7: The SOC content … late seasons, respectively.; The authors need to clearly describe what the average is. Otherwise, four number of percent values (T8 and T1 in early season, T8 and T5 in early season, T8 and T1 in late season, T8 and T5 in late season) must be listed. Before that, The authors’ intent of explaining the difference between T8 and T1 here is not clear, since T1 should be compared to T4 first. (The same is true for Page 7: The total N content …., and Page 7: The concentration of NO3--N ….)
Page 7: The total N content … amended with biochar.; There was no significant difference in TN in some case (T1 and T2 in early season) in Table 2. Even though there are combinations with no significant differences, it is not appropriate to ignore them and describe them as in the text.
Page 7: The total N content … them in late season.; The authors should clearly describe what the total N content in T3, T4, T6 and T7 were higher than. I inferred from the preamble that it meant higher than T1 and T5. But if that is correct, then the second half (however, no significant … in late season) is not appropriate because no significant difference in T3, T6, and T5 in late season in Table 2.
Page 7: The concentration of NH4+-N … 25.25 mg NH4+-N kg-1).; The authors need to clearly describe what the two values of average are. If the first value (28.02) is the average of T2-4 and T6-7 in early season and T2-4 and T6-7 in late season, The authors’ intent of explaining the average is not clear because the amount of urea added was different between T2-4 and T6-7. The same is true for the second value (25.25) if it is the average of T1 and T5 in early and late seasons.
Page 7: Moreover, no significant … in the late season.; the sentence is correct, but I did not understand why it was limited to only the late season because no significant difference between with and without biochar application in early season in Table 2.
Page 7: The nitrification rate … of biochar application.; The authors need to explain from which data they calculated the nitrification rate and the value.
Page 7: The concentration of NO3--N … differences were significant.; There was no significant difference in T8 and T1 in late season in Table 2.
Page 7: The concentration of NO3--N … 11.73 mg kg-1) in T7.; The authors should clearly describe what the values in T8 and T7 were higher than. I did not understand why the authors focused on these two values in different season.
Page 7: Although, no significant … T4, T7, and T8.; I did not understand why the authors mentioned that there were no significant differences in these four values, even though there were other values that were not significantly different (T2 in early season, T5 and T6 in late season).
Page 8: The activity of … application;; Regarding dehydrogenase activity, there was no significant difference between T1, T3 andT4 in late season in Fig 3B. Regarding chitinase activity shown in Fig 3B, there was no significant difference between T1, T2 and T3 in early season and between T1, T2 and T4 in late season. It is not clear from which data this second half-sentence (however … application) indicates.
Page 8: the exception was catalase … in both seasons.; There was no significant difference between in all situations in Fig 2B.
Page 8: The urease activity … T8 than in T1;; There was no significant difference between in T1, T3, T7, and T8 in late season in Fig 2A.
Page 8: In the early … T8 than in T1 (Fig. 2a).; The authors’ intent of calculating the average of 3 situations is not clear because the condition of each situation (amount of urea and biochar) was different.
Page 8: Catalase activity was … sole urea was applied; There was no significant difference between in all situations in both seasons in Fig 2B.
Page 8: In comparison to T1, … 54.29% lower in T4.; The authors’ intent of showing the percentage between T1 and T4 is not clear because there was no significant difference between T1 and T4.
Page 8: When biochar was not … to the early season.; There was no significant difference between in all situations in both seasons in Fig 2B.
Page 8: Catalase activity was … T1 and T5, respectively.; The authors’ intent of showing the percentage between T6 with T1 and T5 is not clear because there was no significant difference between T1, T5 and T6.
It is difficult to comment individually on the content after this.
2. In the discussion chapter (page 14-), the discussion does not make sense because the results are interpreted without the existence of significant differences. Some of them are as follows.
Page 14: pH and concentrations of … the other treatments.; There are combinations with no significant difference between T3, T4, T7, T8 (20 and 30 t B ha-1, applied with 135 or 180 kg N ha-1) and the other treatments (T1, T2, T5, and T8) in Table 2.
Page 15: In our study, soil… and late seasons.; There are combinations (many cases) with no significant difference between biochar-amended soils and no-amended soils in Fig2 and 3.
Page 15: In our study, there … and chitinase activities.; I couldn't find where to find data showing positive interaction. At least, many cases did not indicate no significant difference in Fig 2 and 3.
Page 15: In our study, soil … at 135 kg ha-1 and 180 kg ha-1;; In many cases, there was no significant difference between biochar applied soils and non-applied soils.
Finally, I apologize if my negative comments offend you (the authors). If I did not read accurately your manuscript and you do not agree with my comments (especially, presence of significant differences and interpretation of results), please tell the Editor.
I hope these comments will be helpful.
Author Response
Manuscript ID: microorganisms-2187338
Type of manuscript: Article
Title: Partial substitution of biochar with urea increases enzymes activity, the abundance of ammonia-nitrite oxidizers, and nitrogen uptake in a double-cropping rice system
Authors: Saif Ullah *, Izhar Ali, Mei Yang, Quan Zhao, Anas Iqbal, Xiaoyan Wu, Shakeel Ahmad, Ihsan Muhammad, Abdullah Khan, Muhammad Adnan, Pengli Yuan, Ligeng Jiang
Submitted to section: Microbial Biotechnology,
General and Major Comments
Although I understand that the authors revised their manuscript in response to review comments, I am afraid that this manuscript is not acceptable for the following reasons. Perhaps it is because my comments were not detailed or my English was not clear, but the authors did not properly revise in response to my previous comments.
Respected Reviewer,
Response: We are very thankful that you provided your valuable suggestions and gave us the opportunity to revise our manuscript. We have kindly asked the authors to revise the manuscript. Our point by point response to all the comments and suggestions is given below. Moreover, for better understanding, the changes are marked with track changes in the revised manuscript. We hope that you will find the revised manuscript acceptable for publication in Microorganisms.
- The interpretation of the results is still not objective or scientific. Especially, it is not appropriate for a scientific paper to state that the results were higher or lower, even though there is no significant difference. If there was no significant difference, then it cannot be said to “be higher”. If there were conditions with no significant differences, we cannot ignore them and say that “they were higher”. It should be clearly stated under which conditions there was a significant difference.
In the manuscript, even in the absence of significant differences, the authors stated that the biochar made some parameters higher. The authors also expanded on the conditions where there were some significant differences and stated that biochar made the parameters higher. Additionally, explanation of results was unclear in some areas. The presentation of averages or percentages of values for different conditions is either inappropriate or unclear in intent. Examples in page 7 to 8 are listed below.
Explanatory note: We have thoroughly revised all the results according to your suggestions and the style of explanation in the revised manuscript. Moreover, the results are based on eight treatments and two seasons, so it is hard to compare each treatment with another and describe the results individually as it will create ambiguity for the readers. Although we have tried our best to explain and interpret the results as you directed in the review, By "average," we mean the average of treatments in either season or either treatment, i.e., biochar or urea.
Page 7: pH, SOC, TN, and … in unamended soil.; There was no significant difference in TN in some cases (T1 and T2 in early season, T5 and T6 in late season), in NH4+-N in most cases, in NO3--N in some cases (T5 and T6 in early season, almost all combinations in late season) in Table 2. It is not appropriate to describe that these parameters were higher in the biochar-amended soil, by focusing only on the areas where there were significant differences.
Response: Thank you. We have revised the results of each parameter according to the data set and changes caused by the treatments.
Page 7: The SOC content … late seasons, respectively.; The authors need to clearly describe what the average is. Otherwise, four number of percent values (T8 and T1 in early season, T8 and T5 in early season, T8 and T1 in late season, T8 and T5 in late season) must be listed. Before that, The authors’ intent of explaining the difference between T8 and T1 here is not clear, since T1 should be compared to T4 first. (The same is true for Page 7: The total N content …., and Page 7: The concentration of NO3--N ….)
Response: Thank you. We made significant changes to the results section and compared treatments in low and high dose urea applications, namely T4 with sole urea, T1 and T8, with T5, and then among the remaining treatments.
Page 7: The total N content … amended with biochar.; There was no significant difference in TN in some case (T1 and T2 in early season) in Table 2. Even though there are combinations with no significant differences, it is not appropriate to ignore them and describe them as in the text.
Response: Thank you. In the submitted manuscript in round 2 revision, we resulted it a significant difference on the basis of the data sets for both seasons. In your question, we suspect that this may cause confusion in understanding. Therefore, we have changed it. Please see the revised version.
Page 7: The total N content … them in late season.; The authors should clearly describe what the total N content in T3, T4, T6 and T7 were higher than. I inferred from the preamble that it meant higher than T1 and T5. But if that is correct, then the second half (however, no significant … in late season) is not appropriate because no significant difference in T3, T6, and T5 in late season in Table 2.
Response: Thank you. In accordance with your comments, we have revised it.
Page 7: The concentration of NH4+-N … 25.25 mg NH4+-N kg-1).; The authors need to clearly describe what the two values of average are. If the first value (28.02) is the average of T2-4 and T6-7 in early season and T2-4 and T6-7 in late season, The authors’ intent of explaining the average is not clear because the amount of urea added was different between T2-4 and T6-7. The same is true for the second value (25.25) if it is the average of T1 and T5 in early and late seasons.
Response: We are sorry for this inconvenience. We revised the results and calculated the average separately for low and high dose urea treatments, as well as treatments applied without biochar, in accordance with your instructions.
Page 7: Moreover, no significant … in the late season.; the sentence is correct, but I did not understand why it was limited to only the late season because no significant difference between with and without biochar application in early season in Table 2.
Response: Thank you. We have re-explained it according to both the early and late seasons.
Page 7: The nitrification rate … of biochar application.; The authors need to explain from which data they calculated the nitrification rate and the value.
Response: Thank you. We have revised it according to both the early and late seasons.
Page 7: The concentration of NO3--N … differences were significant.; There was no significant difference in T8 and T1 in late season in Table 2.
Response: Thank you. We have cross-checked and revised it according to the data set given in table 2.
Page 7: The concentration of NO3--N … 11.73 mg kg-1) in T7.; The authors should clearly describe what the values in T8 and T7 were higher than. I did not understand why the authors focused on these two values in different season.
Response: Thank you. As these data were higher among treatments, that’s why we have taken the higher values for comparison; however, in accordance with your comments, we have revised it.
Page 7: Although, no significant … T4, T7, and T8.; I did not understand why the authors mentioned that there were no significant differences in these four values, even though there were other values that were not significantly different (T2 in early season, T5 and T6 in late season).
Response: Thank you. We have revised it according to both the early and late seasons and made a comparison of treatments.
Page 8: The activity of … application;; Regarding dehydrogenase activity, there was no significant difference between T1, T3 andT4 in late season in Fig 3B. Regarding chitinase activity shown in Fig 3B, there was no significant difference between T1, T2 and T3 in early season and between T1, T2 and T4 in late season. It is not clear from which data this second half-sentence (however … application) indicates.
Response: Thank you. Our intent was to describe it generally with an overall effect, although this may not sound authentic, which is why we have revised it in accordance with your reservations.
Page 8: the exception was catalase … in both seasons.; There was no significant difference between in all situations in Fig 2B.
Response: Thank you. Yes, we agree with you, but in the sentence we emphasized that treatments containing biochar induced enzyme activity other than catalase activity; however, we did not mean that the effect was significant. Therefore, to avoid any ambiguity, we have revised our statement.
Page 8: The urease activity … T8 than in T1;; There was no significant difference between in T1, T3, T7, and T8 in late season in Fig 2A.
Response: Thank you. We have revised it.
Page 8: In the early … T8 than in T1 (Fig. 2a).; The authors’ intent of calculating the average of 3 situations is not clear because the condition of each situation (amount of urea and biochar) was different.
Response: Thank you. We revised the results and calculated the average separately for low and high dose urea treatments, as well as treatments applied without biochar, in accordance with your instructions.
Page 8: Catalase activity was … sole urea was applied; There was no significant difference between in all situations in both seasons in Fig 2B.
Response: Thank you. Our intent in the description was that biochar-containing treatments resulted in minimum activity; however, for more clarification, we have revised it.
Page 8: In comparison to T1, … 54.29% lower in T4.; The authors’ intent of showing the percentage between T1 and T4 is not clear because there was no significant difference between T1 and T4.
Response: Thank you. We have corrected it.
Page 8: When biochar was not … to the early season.; There was no significant difference between in all situations in both seasons in Fig 2B.
Response: Thank you. We have revised it.
Page 8: Catalase activity was … T1 and T5, respectively.; The authors’ intent of showing the percentage between T6 with T1 and T5 is not clear because there was no significant difference between T1, T5 and T6.
Response: Thank you. We have revised it.
It is difficult to comment individually on the content after this.
Response: We are happy for your critical check. In accordance with your suggestion and reservation, we have thoroughly revised the results.
- In the discussion chapter (page 14-), the discussion does not make sense because the results are interpreted without the existence of significant differences. Some of them are as follows.
Response: Thank you. We have revised the discussion in accordance with the revised results.
Page 14: pH and concentrations of … the other treatments.; There are combinations with no significant difference between T3, T4, T7, T8 (20 and 30 t B ha-1, applied with 135 or 180 kg N ha-1) and the other treatments (T1, T2, T5, and T8) in Table 2.
Response: Thank you. We have revised the discussion in accordance with the revised results.
Page 15: In our study, soil… and late seasons.; There are combinations (many cases) with no significant difference between biochar-amended soils and no-amended soils in Fig2 and 3.
Response: Thank you. We have revised it.
Page 15: In our study, there … and chitinase activities.; I couldn't find where to find data showing positive interaction. At least, many cases did not indicate no significant difference in Fig 2 and 3.
Response: Thank you. We have revised it.
Page 15: In our study, soil … at 135 kg ha-1 and 180 kg ha-1;; In many cases, there was no significant difference between biochar applied soils and non-applied soils.
Response: Thank you. We have revised it and described it individually with each case.